# Inhibition of complement activation, myeloperoxidase, NET formation and oxidant activity by PIC1 peptide variants

**Pamela S. Hair[1], Adrianne I. Enos[1], Neel K. Krishna[1,2], Kenji M. Cunnion[1,2,3,4]***

**1** Department of Pediatrics, Eastern Virginia Medical School, Norfolk, VA, United States of America,
**2** Department of Microbiology and Molecular Cell Biology, Eastern Virginia Medical School, Norfolk, VA, United States of America, **3** Children's Specialty Group, Norfolk, VA, United States of America, **4** Children's Hospital of The King's Daughters, Norfolk, VA, United States of America

\* cunniokm@evms.edu

**Data Availability Statement:** All relevant data are within the manuscript and its Supporting Information files.

**Funding:** This work was supported by NIH grant R21 AI135222. The funder did not play a role in the

## Abstract

### Background

A product of rational molecular design, PA-dPEG24 is the lead derivative of the PIC1 family of peptides with multiple functional abilities including classical complement pathway inhibition, myeloperoxidase inhibition, NET inhibition and antioxidant activity. PA-dPEG24 is composed of a sequence of 15 amino acid, IALILEPICCQERAA, and contains a monodisperse 24-mer PEGylated moiety at its C terminus to increase aqueous solubility. Here we explore a sarcosine substitution scan of the PA peptide to evaluate impacts on solubility in the absence of PEGylation and functional characteristics.

### Methods

Sixteen sarcosine substitution variants were synthesized and evaluated for solubility in water. Aqueous soluble variants were then tested in standard complement, myeloperoxidase, NET formation and antioxidant capacity assays.

### Results

Six sarcosine substitution variants were aqueous soluble without requiring PEGylation. Substitution with sarcosine of the isoleucine at position eight yielded a soluble peptide that surpassed the parent molecule for complement inhibition and myeloperoxidase inhibition. Substitution with sarcosine of the cysteine at position nine improved solubility, but did not otherwise change the functional characteristics compared with the parent compound. However, replacement of both vicinal cysteine residues at positions 9 and 10 with a single sarcosine residue reduced functional activity in most of the assays tested.

### Conclusions

Several of the sarcosine PIC1 variant substitutions synthesized yielded improved solubility as well as a number of unanticipated structure-function findings that provide new insights. Several sarcosine substitution variants demonstrate increased potency over the parent

study design, data collection and analysis, decision to publish, or preparation of the manuscript. https://grants.nih.gov/grants/funding/r21.htm

**Competing interests:** I have read the journal's policy and the authors of this manuscript have the following competing interests: Kerry Cunnion also serves as Chief Medical Officer for ReAlta Life Sciences. Neel Krishna also serves as Chief Science Officer for ReAlta Life Sciences. This does not alter our adherence to PLoS ONE policies on sharing data and materials.

**Abbreviations:** αhistone, anti-histone antibody; αNE, anti-neutrophil elastase; CRE, copper reducing equivalents; EA, erythrocytes; HAT, hydrogen atom transfer; HNP-1, human neutrophil defensin type 1; IC, immune complexes; MPO, myeloperoxidase; NET, neutrophil extracellular trap; NHS, normal human serum; PBST, phosphate buffered saline + 0.1% Tween; PIC1, PA-dPEG24; SEM, standard error; SET, singe electron transfer; TAC, Total Antioxidant Capacity; TMB, tetramethylbenzidine.

peptide suggesting enhanced therapeutic potential for inflammatory disease processes involving complement, myeloperoxidase, NETs or oxidant stress.

## Introduction

Human astroviruses belong to a family of non-enveloped, icosahedral RNA viruses that are an endemic, world-wide pathogen causing acute gastroenteritis in human infants [1]. Curiously, unlike calicivirus and rotavirus that cause severe acute disease, astrovirus gastroenteritis is non-inflammatory [2]. To understand the blunted inflammatory response to this pathogen, our laboratories demonstrated that the 787 amino acid residue coat protein that forms the astrovirus capsid inhibited activation of the classical pathway of complement [3]. Complement is an innate immune response of humans that is characterized by a robust inflammatory response to pathogens [4]. Analysis of the amino acid sequence of the astrovirus coat protein identified a region with loose homology to human neutrophil defensin type 1 (HNP-1) and led to development of short peptides that retained the ability to inhibit the classical pathway of complement [5]. Subsequent extensive rational drug design yielded a multifunctional 15 amino acid peptide (IALILEPICCQERAA) with a 24mer monodisperse PEG moiety on the C terminus of the peptide, (PA-dPEG24 or PIC1) to improve solubility [6]. This sequence no longer possesses significant homology with known naturally occurring protein or peptide sequences.

Previous publications have described the ability of PA-dPEG24 to bind C1q and inhibit the enzymatic activity of C1s, blocking activation of the classical complement pathway [6]. In addition, PA-dPEG24 inhibits the peroxidase activity of heme-based molecules including myeloperoxidase (MPO) [7], hemoglobin and myoglobin [8], inhibits neutrophils from undergoing NETosis when stimulated by PMA or immune complexes [9] and inhibits oxidation and reduces molecules via the singe electron transfer (SET) and hydrogen atom transfer (HAT) mechanisms [10]. As PA-dPEG24 retains defensin-like characteristics being amphiphilic and cysteine-rich, we have additionally shown it can bind the surface of *S. aureus* and *K. pneumoniae* and has antimicrobial activity against these bacteria and *N. meningitidis* [11]. In the same manuscript, we also described the complement inhibitory and antimicrobial properties of a few sarcosine substitution variants of the 15 amino acid base peptide. Sarcosine substitution variants were developed in order to improve solubility of the IALILEPICCQERAA peptide such that the dPEG24 tail would no longer be necessary. Here we delve into how the sarcosine substitutions alter the solubility and biological functions and of the peptide.

## Materials and methods

### Ethics statement

Blood from healthy donors was obtained with written consent under an Eastern Virginia Medical School IRB approved protocol, 02-06-EX 0216. Blood was used for the preparation of reagents: purified platelets, erythrocytes and neutrophils.

### Reagents

PA-dPEG24 (IALILEPICCQERAA-dPEG24 or PIC1) was manufactured by PolyPeptide Group (San Diego, CA) to ≥ 95% purity verified by HPLC and mass spectrometry analysis. Lyophilized PA-dPEG24 was solubilized in 0.05 M Histidine buffer and pH adjusted to 6.7.

**Table 1. Peptide designations and sequences.**

| Name | Sequence | H2O soluble |
|---|---|---|
| PA-dPEG24 (PIC1) | H2N-IALILEPICCQERAA-dPEG24 | N |
| PA | H2N-IALILEPICCQERAA-OH | N |
| I1 | H2N-(Sar)ALILEPICCQERAA-OH | N |
| A2 | H2N-I(Sar)LILEPICCQERAA-OH | Y |
| L3 | H2N-IA(Sar)ILEPICCQERAA-OH | Y |
| I4 | H2N-IAL(Sar)LEPICCQERAA-OH | Y |
| L5 | H2N-IALI(Sar)EPICCQERAA-OH | Y |
| E6 | H2N-IALIL(Sar)PICCQERAA-OH | N |
| P7 | H2N-IALILE(Sar)ICCQERAA-OH | N |
| I8 | H2N-IALILEP(Sar)CCQERAA-OH | Y |
| C9 | H2N-IALILEPI(Sar)CQERAA-OH | Y |
| C10 | H2N-IALILEPIC(Sar)QERAA-OH | N |
| C9,10 | H2N-IALILEPI(Sar)QERAA-OH | Y |
| Q11 | H2N-IALILEPICC(Sar)ERAA-OH | N |
| E13 | H2N-IALILEPICCQ(Sar)RAA-OH | N |
| R14 | H2N-IALILEPICCQE(Sar)AA-OH | N |
| A15 | H2N-IALILEPICCQER(Sar)A-OH | N |
| A16 | H2N-IALILEPICCQERA(Sar)-OH | N |

Sarcosine substitution derivative peptides and the base peptide IALILEPICCQERAA (PA) (Table 1) were synthesized by New England Peptide (Gardner, MA) to >90% purity. Sarcosine variants and PEG were dissolved in water and the pH was adjusted with NaOH. PA was dissolved in DMSO and then brought up to the final concentration with water resulting in 30% DMSO and pH adjusted. Antibody sensitized sheep erythrocytes (EA), purified C1q and factor B-depleted human sera were purchased from Complement Technology (Tyler, TX). Purified myeloperoxidase was purchased from Lee BioSolutions (Maryland Heights, MO) and tetramethylbenzidine (TMB) and PicoGreen were purchased from Thermo Fisher (Waltham MA).

## Buffers

Complement permissive GVBS$^{++}$ buffer is veronal buffered saline with 0.1% gelatin, 0.15 mM CaCl$_2$, and 1 mM MgCl$_2$ [12]. Complement inhibitory buffer GVBS$^{--}$ is a veronal-buffered saline with 0.1% gelatin and 10mM EDTA.

## Pooled Normal Human Serum (NHS)

Pooled normal human serum (NHS) was prepared as previously described [12]. Briefly, blood from at least 4 healthy human donors is collected in Vacutainer tubes without additives (red top). The blood sits for 30 minutes at room temperature and 2 hours on ice to clot the blood, separating the serum. The sera are then pooled, aliquoted and frozen at -80ºC.

## Hemolytic assays of complement activity

For hemolytic complement assays, human red blood cells (RBCs) from type AB donors were purified, washed and standardized to $1 \times 10^9$ cells/ml, as previously described [13]. Human sera from type O donors at a 20% final concentration was combined with 1 mM PIC1 or sarcosine variant peptides and the volume was brought up to 0.15 ml with GVBS$^{++}$ and 0.5 ml RBCs. For factor B-depleted sera hemolytic assays, a final of 0.005% factor B depleted sera was

incubated with 1 mM PIC1 or sarcosine variant peptides with 0.1 ml antibody-sensitized sheep red blood cells (EA) in a final volume of 0.75 ml GVBS$^{++}$. The samples were incubated for 1 hour at 37ºC and then 1.0 ml of GVBS$^{--}$was added to the factor B-depleted samples to stop the reaction. The samples were spun at 3,000 rpm for 5 minutes and the supernatant was collected and read at 412 nm. Values are represented as a percent of the positive control, which consists of human O sera and AB red blood cells in GVBS$^{++}$ buffer.

## MPO activity assay

PIC1 and sarcosine variants were diluted to 25 mg/ml and then serially titrated in a 96 well plate at a volume of 0.02 ml. Myeloperoxidase (MPO) was diluted to 20 µg/ml and 0.02 ml was added to the titrated peptides. TMB (3,3′,5,5′-tetramethylbenzidine) (0.1 ml) was added to each well for 2 minutes, followed by 0.1 ml of 2.5 N $H_2SO_4$ for another 2 minutes, and then read on a 96 well plate reader (BioTek) at 450 nm.

## C1q and MPO binding assays

An Immunlon-2 HB ELISA plate was coated with 1 µg/ml C1q or MPO in bicarbonate buffer overnight at 4ºC. The plates were washed with PBST (phosphate buffered saline + 0.1% Tween) and then blocked with 1% gelatin/PBS for 2 hours at room temperature. After washing, the plates were incubated with PA or sarcosine variant peptides starting at 2.5 mg/ml and then serially diluted in 1% gelatin/PBS for 1 hour at room temperature. After washing, the plates were probed with rabbit anti-PA (developed with Cocalico Biologicals, Reamstown, PA) at 1:1000 in 1% gelatin/PBS for 1 hour, room temperature, followed by a goat anti-rabbit HRP (Sigma Aldrich, St Louis, MO) at 1:1,000 in 1% gelatin/PBS for 1 hour at room temperature with a washing step in between. Wells were developed using TMB substrate solution, stopped using 1 N $H_2SO_4$, and read at 450 nm in a plate reader.

## MPO heme ring oxidative degradation

In a 96 well plate, 0.025 ml of 1.7 mg/ml MPO was combined with 0.00125 ml of 0.5% $H_2O_2$ and 3 mM PIC1 or sarcosine variant peptides in a total volume of 0.125 ml PBS and left to incubate at room temperature for 2 minutes. Using a 96 well plate reader, the wells were scanned for absorbance from 300–550 nm to generate curves reflecting the iron state in the MPO heme ring, as previously reported [14].

## Total Antioxidant Capacity assay

The TAC (Total Antioxidant Capacity) Assay (Cell Biolabs, Inc, San Diego, CA) was used to measure the antioxidant capacity of the sarcosine variants based on the reduction of copper (II) to copper (I). The kit protocol was performed as recommended.

## NETosis assay

NETosis assays with immune complexes were performed as previously described [9]. Briefly, normal human serum was stimulated with ovalbumin-antiovalbumin immune complexes in GVBS$^{++}$ for 30 minutes at 37ºC. This mixture along with 0.05% $H_2O_2$ was then added to purified human neutrophils resuspended in RPMI with or without sarcosine variants (2 mM), allowing NETosis to occur. Quantitation of free DNA released from the neutrophils was performed using PicoGreen. Slides were stained with DAPI (Southern Biotech, Birmingham, AL) and the following antibodies, mouse anti-elastase (Invitrogen, Carlsbad, CA) was used with the secondary goat anti-mouse Alexa Fluor 568 (Novus Biologicals, Centennial, CO) and

rabbit anti-histone H3 (Abcam, Cambridge MA) with goat anti-rabbit Alexa Fluor 488 (Novus Biologicals) to visualize formation of NETs which were imaged with fluorescence microscopy with an Olympus BX53 microscope.

## Statistical analysis

Quantitative data were analyzed determining means, standard error (SEM), and Student's t-test [15] using Excel (Microsoft, Redmond, WA).

## Results

### Peptide solubility

To evaluate the influence of sarcosine residues on the solubility and biological function of the base peptide IALILEPICCQERAA (PA), peptide derivatives were synthesized with sarcosine residues substituted at all 15 positions, included a peptide in which the vicinal cysteines at positions 9 and 10 (C9,C10) were replaced with a single sarcosine residue (Table 1). Substitution of sarcosine at positions A2, L3, I4, L5, I8, C9 and C9,10 resulted in peptides soluble in water (Table 1). Due to their enhanced solubility in the absence of PEGylation, these peptides were chosen for further evaluation of the various biological activities identified for the lead peptide PA-dPEG24 or PIC1.

### Complement inhibition

In order to evaluate the extent to which the peptide variants inhibit antibody-initiated complement activation, we tested them in two hemolytic assays including an ABO incompatibility *ex vivo* assay (Fig 1A) and a classical pathway CH50-type assay in factor B-depleted sera (Fig 1B). In the ABO incompatibility hemolytic assay, purified erythrocytes from a 'type AB+' donor are incubated with sera from a 'type O' subject containing anti-A and anti-B antibodies; peptides were tested at 1.8 mM. Variants A2, I4, I8 and C9 each inhibited ABO incompatible hemolysis to a greater extent than did the PA-dPEG24 (PIC1) parent compound on an equimolar basis ($P < 0.015$). The I8 variant decreased ABO hemolysis 53% ($P < 0.002$) more than PA-dPEG24. The C9,10 variant shows minimal inhibition of ABO hemolysis. We then performed a CH50-type hemolytic assay, with antibody-sensitized sheep erythrocytes, isolating the classical pathway by utilizing factor B-depleted sera; peptides were tested at 0.4 mM. In this assay the I8 variant demonstrated superior activity inhibiting hemolysis 75% ($P < 0.001$) more than PA-dPEG24. Other peptides demonstrated similar inhibition of the classical complement pathway compared with PA-dPEG24 with the exception of C9,10, which again showed minimal activity.

We then tested peptide variant binding to C1q in an ELISA-type assay in which the C1q is used as the capture substrate. Binding curves for each peptide is shown in Fig 1C, from which half-maximal binding concentrations were calculated (Fig 1D). These binding curves and half-maximal binding calculations demonstrate that I8 and PA, the parent peptide sequence, yield superior binding to C1q compared with the other peptides. The PA variant has poor aqueous solubility, such that it needs to be initially solubilized in DMSO and then diluted into an aqueous buffer. Higher concentrations of DMSO interfere with the detecting reagents resulting in a partial binding curve. The superior C1q binding of I8 correlates with superior inhibition of complement mediated hemolysis. Overall, the I8 variant shows superior inhibition of antibody-initiated complement activation and hemolysis compared with the parent compound and other peptide variants.

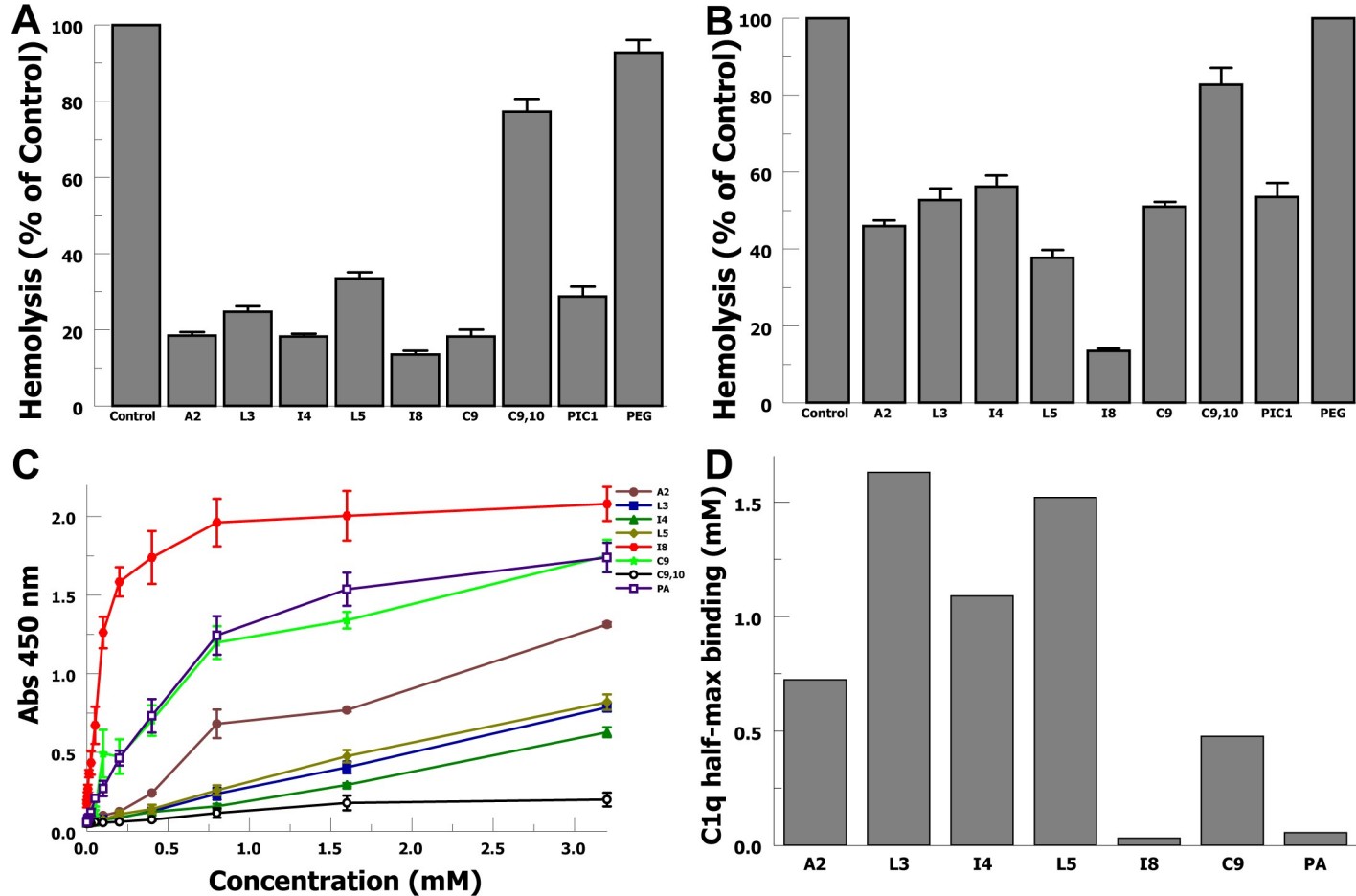

**Fig 1. Sarcosine variant inhibition of complement activation in hemolytic assays and C1q binding.** A) Inhibition of ABO incompatibility hemolysis in a CH50-type assay. Peptides are at a final concentration of 1.8 mM. PIC1 denotes PA-dPEG24. Data are the means of n = 4 independent experiments + SEM. B) Inhibition of classical complement pathway-mediated hemolysis in factor B-depleted sera in a CH50-type assay. Peptides are at a final concentration of 0.4 mM. Data are the means of n = 4 independent experiments + SEM. C) Binding of increasing concentrations of sarcosine variants to purified C1q in an ELISA-type assay. Data are the means of n = 3 independent experiments ± SEM. D) Half-maximal binding concentrations were calculated for each peptide's binding curve.

## Myeloperoxidase inhibition and binding

Next we tested inhibition of MPO activity in a TMB-based in vitro assay, as previously described for PA-dPEG24 [7]. In this assay, the variants were tested for MPO inhibition over a range of concentrations (Fig 2A). Strong MPO inhibition was found for all variants with the exception of the no-cysteine variant (C9,10). We calculated half-maximal inhibition values from the dose-response curves for each variant and demonstrated measurable differences in MPO inhibition (Fig 2B). Variant I8 again showed the greatest potency among the different variants.

We then tested the binding of the peptide variants to solid phased MPO in a plate-based assay. The binding curves are shown in Fig 2C with MPO binding identified for all variants except C9,10. Due to the near complete overlay of the I8 and PA curves, the PA curve is not shown in the graph. Half-maximal (Fig 2D) binding concentrations were calculated from these curves. I8 demonstrated superior binding to MPO compared with the other sarcosine variants consistent with the increased MPO inhibition identified by the dose-response data above.

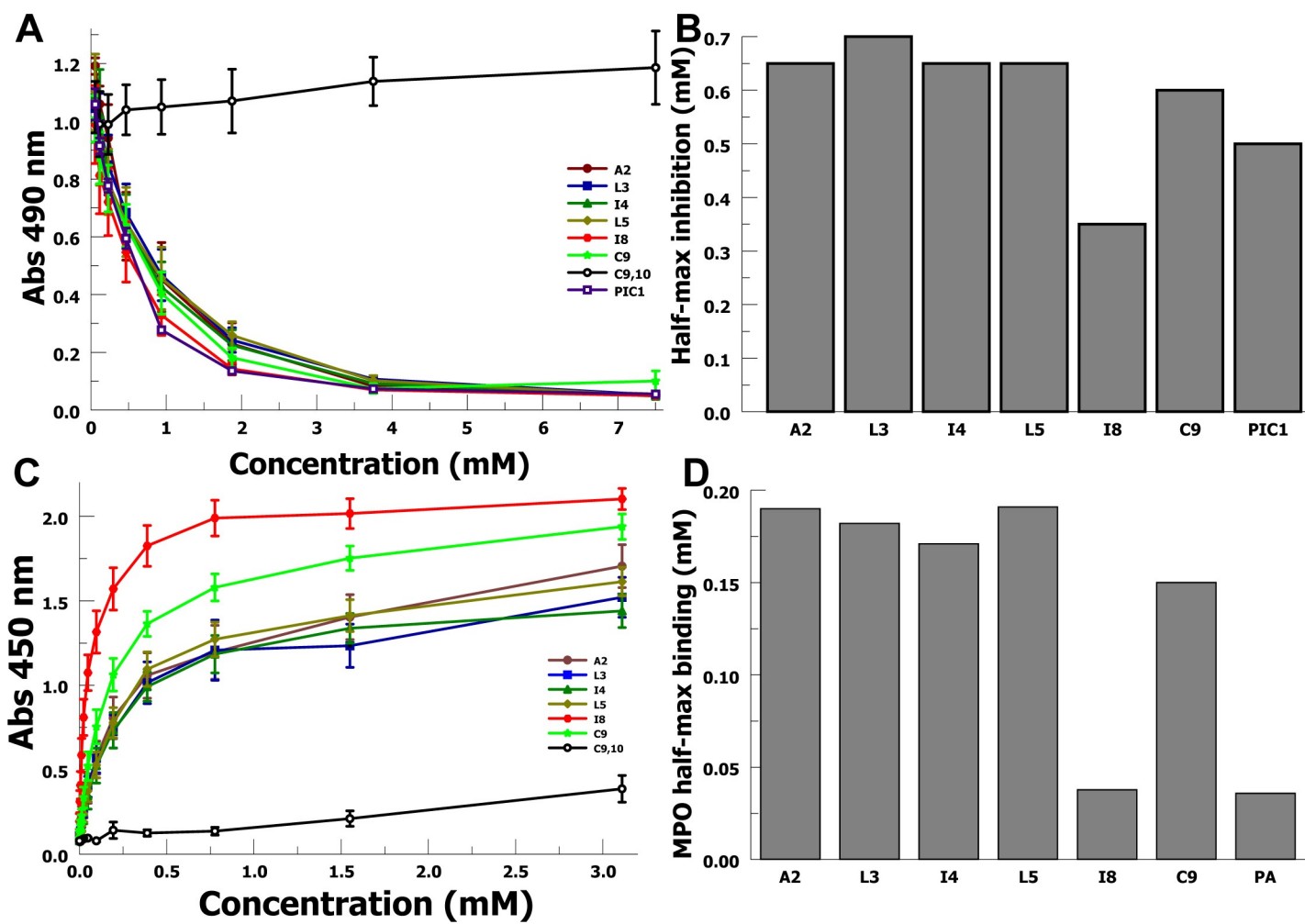

**Fig 2. Sarcosine variant inhibition of MPO peroxidase activity.** A) MPO peroxidase activity was measured in a TMB-based assay for each peptide over a range of concentrations (mM). PIC1 denotes PA-dPEG24. Data are the means of n = 3 independent experiments ± SEM. B) Half-maximal inhibition concentrations were calculated for each peptide's inhibition curve. C) Binding of increasing concentrations of sarcosine variants to purified MPO in an ELISA-type assay. Data are the means of n = 3 independent experiments ± SEM. D) Half-maximal binding concentrations were calculated for each peptide's binding curve.

## Protection of the MPO heme ring from degradation

We then evaluated the ability of the variants to prevent degradation of the heme ring in the MPO molecule. As MPO produces hypochlorous acid in the presence of chloride and hydrogen peroxide, the hypochlorous acid will degrade the heme ring, as previously shown for PA-dPEG24 [8]. This can be evaluated via spectrometric measurements of absorption of wavelengths from 300–550 nm. Absorption measurements were made over this spectrum for all variants (Fig 3A–3H). All peptides showed some inhibition of heme ring degradation with near perfect preservation of the heme ring absorption spectrum for the I8 and C9 variants. The heme ring preservation for these two variants was greater than that for the parent compound, PA-dPEG24. The C9,10 variant also showed inhibition of heme ring degradation, suggesting a mechanism of protection other than inhibition of MPO peroxidase activity, which the C9,10 variant does not possess (Fig 2A).

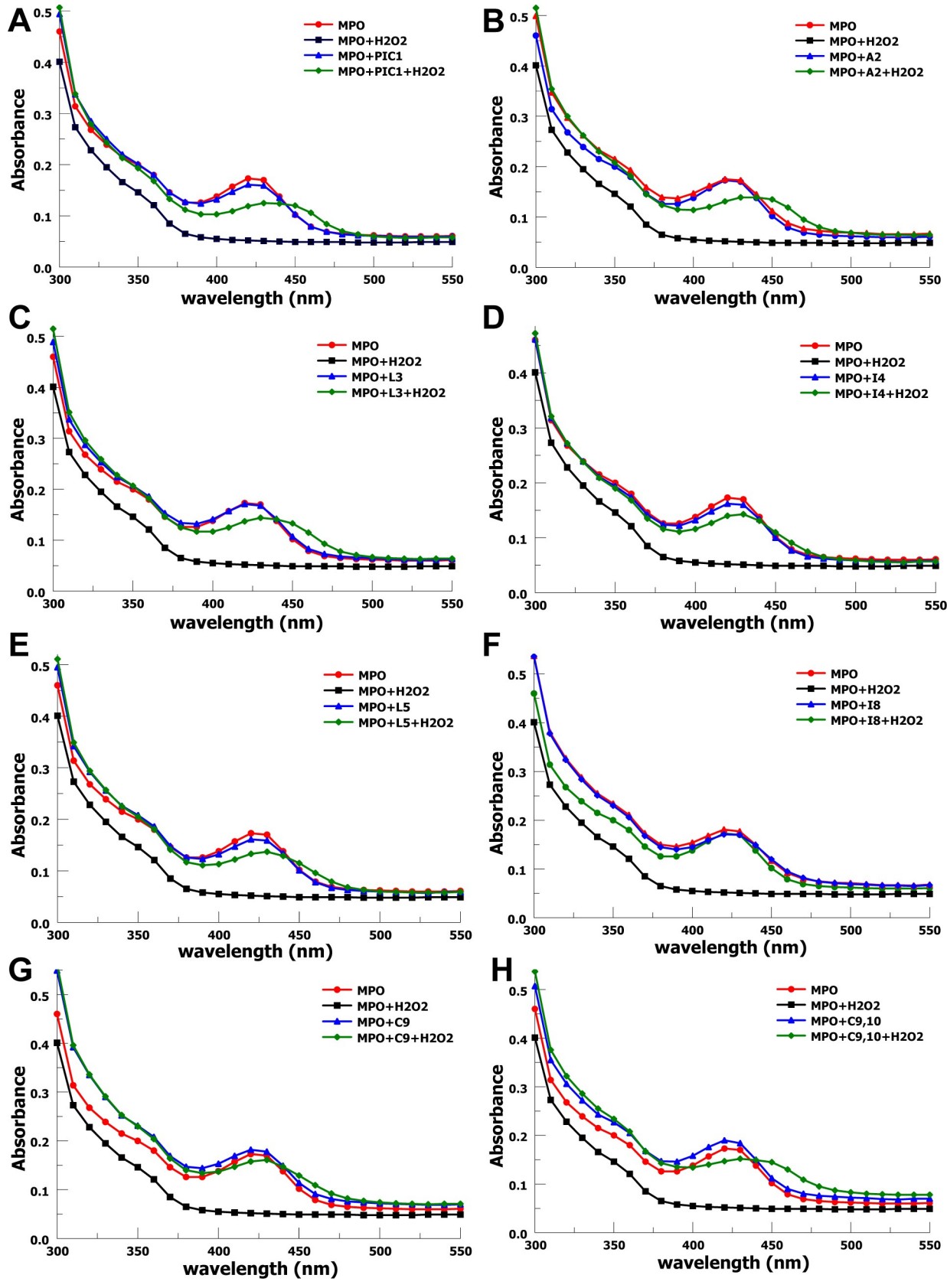

**Fig 3. Sarcosine variant protection of MPO heme ring from oxidative degradation.** Absorbance spectra for the heme ring is shown in the absence of hydrogen peroxide (MPO) in red, in the presence of hydrogen peroxide (MPO+H2O2) in black, in the presence of peptide in blue, and in the presence of hydrogen peroxide and peptide together in green. A) PIC1 denotes PA-dPEG24. B-H) Peptides A2 – C9,10 are denoted in the legends. Peptides were tested at a concentration of 3.0 mM.

## Antioxidant capacity

We then tested antioxidant capability for the variants in a Total Antioxidant Capacity (TAC) assay, as previously described for PA-dPEG24 [10]. The variants showed differing total antioxidant capacity (Fig 4) with L3, I4 and I8 demonstrating antioxidant capacity approaching that of PA-dPEG24 (PIC1). The C9,10 no-cysteine variant demonstrated no antioxidant capacity, as expected.

## NETosis inhibition

The peptide variants were tested for inhibition of NETosis, which we have previously described for PA-dPEG24 [9]. In this assay purified human neutrophils are stimulated with normal human serum activated with ovalbumin-antiovalbumin immune complexes and hydrogen peroxide. Free DNA expressed from the neutrophils is then measured in a Pico-Green assay. In this assay multiple variants including A2, L3, L5, I8 and C9 demonstrated similar ability to inhibit NETosis compared with PA-dPEG24 (Fig 5 graph). I4 demonstrated a reduced ability to inhibit NETosis compared with PA-dPEG24 and C9,10 showed only slight inhibition relative to the negative control (buffer). These results show that many of the sarcosine variant peptides are able to inhibit NETosis to baseline values of free DNA from unstimulated neutrophils.

Confirmation of NETosis results with the I8 variant was performed via direct visualization of neutrophil extracellular trap (NET) formation by fluorescence microscopy. Human neutrophils were purified and stimulated on glass slides then stained with DAPI to visualize DNA and probed with anti-neutrophil elastase (αNE) and anti-histone H3 (αhistone), as previously described [9]. Unstimulated neutrophils showed no evidence of NETs (Fig 5 –first row), neutrophils stimulated with ovalbumin-antiovalbumin immune complexes and hydrogen

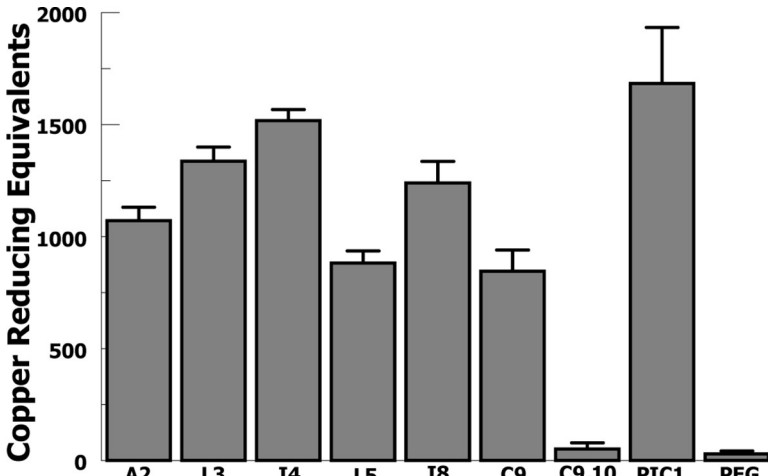

**Fig 4. Sarcosine variant inhibition of oxidant activity in a Total Antioxidant Capacity (TAC) assay.** Antioxidant activity is measured in copper reducing equivalents (CRE). Peptides were tested over a range of concentrations from 0.03–0.25 mM then compared to the standard. Data are the means of n = 3 independent experiments + SEM.

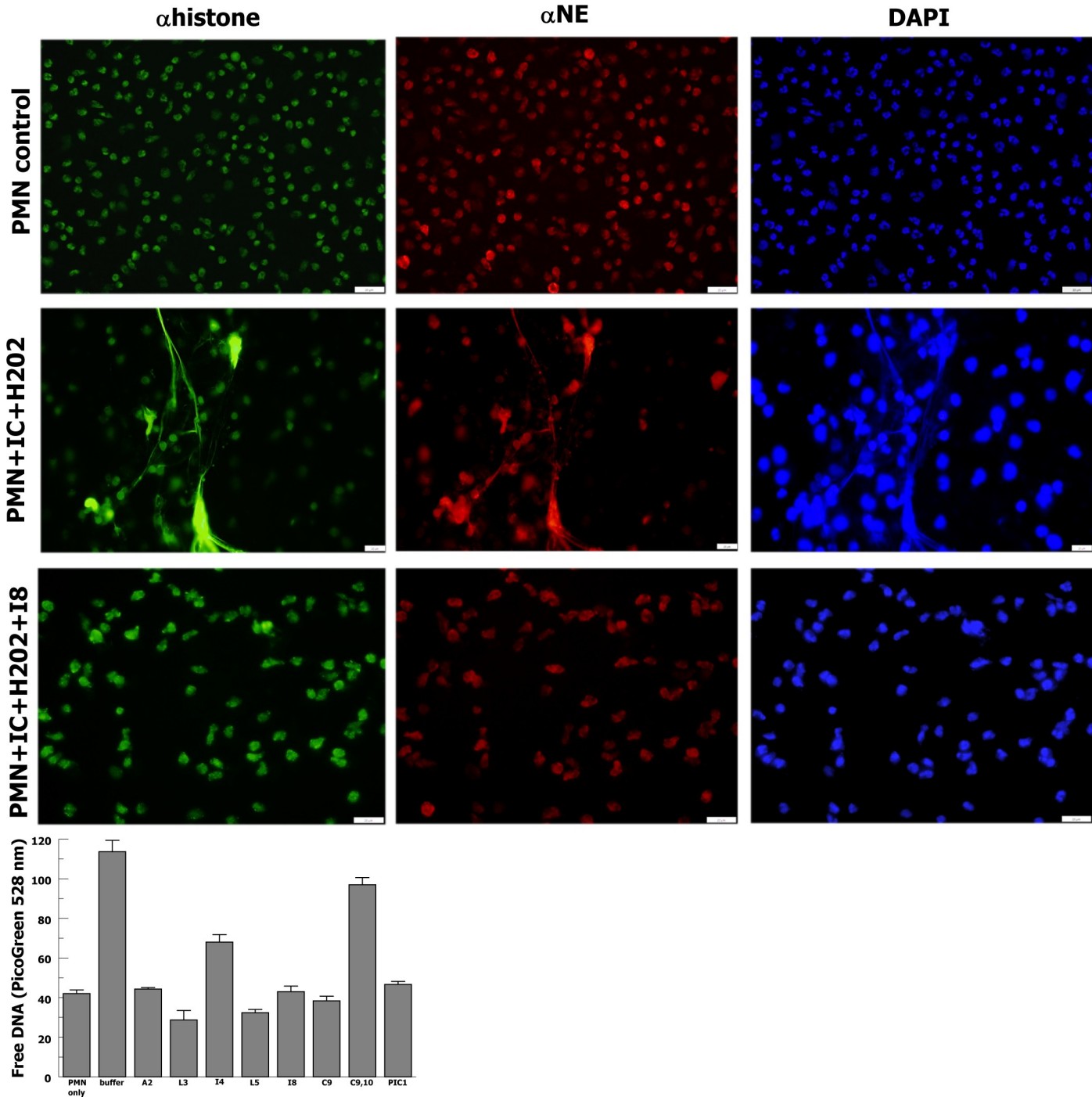

**Fig 5. Sarcosine variant inhibition of NET formation.** Fluorescence microscopy images of NET formation after neutrophil stimulation with immune complexes (IC) and hydrogen peroxide (H2O2) compared with neutrophil only control. Variant I8 inhibition of NET formation is shown in the third row. Histones are probed with anti-histone antibody (αhistone), neutrophil elastase is probed with anti-neutrophil elastase (αNE) and DNA is stained with DAPI. Representative images are shown. The graph shows sarcosine variant inhibition of free DNA release by neutrophils as a marker of NETosis. Purified human neutrophils were stimulated with 2% normal human sera pre-incubated with ovalbumin-antiovalbumin immune complexes, and 0.05% H2O2. Sarcosine peptides and PIC1 were added to the sera to a final concentration of 2 mM. Data are the means of n = 3 independent experiments + SEM.

peroxide showed many NETs (Fig 5 –second row) and neutrophils stimulated with immune complexes in the presence of PA-dPEG24 showed a dramatic reduction in NETs (Fig 5 –third row). Fluorescence microscopy visualization of NETs confirmed the findings seen for quantitative measurements of free DNA.

## Discussion

Here we investigate the functionalities of the aqueous soluble sarcosine substitution derivatives of PA-dPEG24. Our primary goal was to improve the solubility of the peptide IALILEPICC-QERAA by employing sarcosine amino acid substitutions such that the dPEG24 tail would no longer be required and thereby decreasing the size of the molecule by about half.

Individual substitution of sarcosine at each of the 15 amino acids yielded 6 sarcosine variants with good aqueous solubility without PEGylation (Table 1). Sarcosine substitution of the relatively hydrophobic alanine, leucine and isoleucine amino acids at positions 2–5 and position 8 dramatically improved solubility as might be expected from decreasing the overall hydrophobicity of the peptide. Interestingly, sarcosine substitution of cysteine at position 9, but not at position 10, also improved solubility as did substitution of both vicinal cysteines (C9,10) with a single sarcosine. Cysteine residues are uncharged and relatively polar and the observation that solubility is enhanced with sarcosine substitution at positions 9, 9 and 10 but not 10 suggests that the substitution is affecting solubility by altering the conformation of the peptide as opposed to modifying overall hydrophobicity. Sarcosine residues are frequently used in medicinal chemistry due to the favorable solubility profile, reduction in the number of intra- or inter- molecular hydrogen bonds due to absence of the proton from the NH group and potential alteration of neighboring residues due to changes in the $\phi$, $\psi$ torsion angles resulting in increased steric constraints [16].

Due to the multiple functions previously discovered and described for the parent PIC1 molecule, we investigated how the sarcosine substitutions might affect biological activity. The functional data for the sarcosine variants shown in the figures is summarized in Table 2. An unanticipated finding is that the I8 variant is superior to the parent compound, PA-dPEG24, in most of the anti-inflammatory assays tested. The C1q binding for I8 is slightly superior to that of PA-dPEG24 and may be driving enhanced inhibition of C1 activation.

Also unanticipated is that the C9 variant performed almost identically to the parent compound, PA-dPEG24, in terms of complement inhibition, despite the loss of one of the two cysteine amino acids. In the total antioxidant capacity assay, C9 had approximately half of the antioxidant capacity as PA-dPEG24, commensurate with having half as many cysteines. In the complement inhibition assays where no cysteines were present (variant C9,10) most, but not

**Table 2. Summary of peptide properties.**

| Assay | PA-dPEG24 | A2 | L3 | I4 | L5 | I8 | C9 | C9,10 |
|---|---|---|---|---|---|---|---|---|
| ABO C' inhibition | +++ | ++++ | +++ | ++++ | +++ | ++++ | +++ | 0 |
| Classical C' inhibition | ++ | ++ | ++ | ++ | +++ | ++++ | ++ | 0 |
| MPO inhibition | +++ | ++++ | +++ | +++ | +++ | ++++ | +++ | 0 |
| NETosis inhibition | +++ | +++ | ++++ | ++ | ++++ | +++ | +++ | + |
| Heme ring protection | ++ | ++ | ++ | +++ | +++ | ++++ | +++ | ++ |
| Total Antioxidant Cap | ++++ | ++ | +++ | ++++ | ++ | +++ | ++ | 0 |
| Antibacterial activity* | ++ | ++++ | ++++ | ++++ | +++ | ND | ND | ND |

*previously published

ND = not done

all, complement inhibition was lost. Host complement inhibitors are typically cysteine rich [17]. Together these findings suggest that the cysteines contribute to complement inhibition, but do not account for the entirety of complement inhibition for these peptides.

Overall, these studies provide new data on how sarcosine substitution of standard amino acids in a peptide can not only change solubility, but also biological functions in unpredictable ways. Additionally, the roles of the vicinal cysteines for the biological functions of these molecules is also further defined. Sarcosine containing bioactive peptides has also been demonstrated to have increased stability to enzymatic degradation [16]. Future studies will test the most active of these sarcosine variants of PIC1 in animal models to evaluate if the sarcosine variants increase the circulating half-life and biological activity of the molecules as well as their pathophysiological impact in disease models.

Peptides are an active area of research and development with potential applicability for use as therapeutic medications. The human body uses peptides for a number of important functions from endocrine (e.g. insulin) to danger signaling (e.g. C5a). Peptides are naturally broken down in the body by peptidases to constituent amino acids that can be recycled. Peptides are also relatively easily synthesized chemically by methods like Fmoc solid phase protein synthesis. For these reasons, peptides are gaining favor for therapeutic medication development. The new peptides described above are smaller and, in some instances, more potent than the parent molecule demonstrating that peptides are exceptionally amenable to rational drug design methods. The peptides described above have potential therapeutic application for diseases mediated by dysregulation of the complement system, myeloperoxidase, NET formation or oxidant stress.

## Supporting information

**S1 Dataset.**
(XLSX)

## Author Contributions

**Conceptualization:** Kenji M. Cunnion.

**Data curation:** Pamela S. Hair, Adrianne I. Enos, Kenji M. Cunnion.

**Formal analysis:** Pamela S. Hair, Adrianne I. Enos, Kenji M. Cunnion.

**Funding acquisition:** Kenji M. Cunnion.

**Investigation:** Pamela S. Hair, Neel K. Krishna, Kenji M. Cunnion.

**Methodology:** Pamela S. Hair, Kenji M. Cunnion.

**Project administration:** Pamela S. Hair, Neel K. Krishna, Kenji M. Cunnion.

**Resources:** Kenji M. Cunnion.

**Software:** Kenji M. Cunnion.

**Supervision:** Pamela S. Hair, Neel K. Krishna, Kenji M. Cunnion.

**Validation:** Adrianne I. Enos, Neel K. Krishna, Kenji M. Cunnion.

**Visualization:** Kenji M. Cunnion.

**Writing – original draft:** Kenji M. Cunnion.

**Writing – review & editing:** Neel K. Krishna, Kenji M. Cunnion.

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
