## [Decision Letter · Decision Letter 0]

6 Nov 2019

PONE-D-19-26436

Inhibition of complement activation, myeloperoxidase, NET formation and oxidant activity by PIC1 peptide variants.

PLOS ONE

Dear Dr. Cunnion,

Thank you for submitting your manuscript to PLOS ONE. After careful consideration, we feel that it has merit but does not fully meet PLOS ONE’s publication criteria as it currently stands. Therefore, we invite you to submit a revised version of the manuscript that addresses the points raised during the review process.

We would appreciate receiving your revised manuscript by Dec 21 2019 11:59PM. To enhance the reproducibility of your results, we recommend that if applicable you deposit your laboratory protocols in protocols.io, where a protocol can be assigned its own identifier (DOI) such that it can be cited independently in the future. For instructions see: http://journals.plos.org/plosone/s/submission-guidelines#loc-laboratory-protocols

We look forward to receiving your revised manuscript.

Kind regards,

Yan Li

Academic Editor

PLOS ONE

Journal Requirements:

1. Thank you for including your competing interests statement; "I have read the journal's policy and the authors of this manuscript have the following competing interests: Kerry Cunnion also serves as Chief Medical Officer for ReAlta Life Sciences. Neel Krishna also serves as Chief Science Officer for ReAlta Life Sciences."

Reviewers' comments:

Reviewer's Responses to Questions

**Comments to the Author**

1. Is the manuscript technically sound, and do the data support the conclusions?

Reviewer #1: Yes

Reviewer #2: Yes

Reviewer #3: Yes

2. Has the statistical analysis been performed appropriately and rigorously? 

Reviewer #1: Yes

Reviewer #2: Yes

Reviewer #3: Yes

3. Have the authors made all data underlying the findings in their manuscript fully available?

Reviewer #1: Yes

Reviewer #2: Yes

Reviewer #3: Yes

4. Is the manuscript presented in an intelligible fashion and written in standard English?

Reviewer #1: Yes

Reviewer #2: Yes

Reviewer #3: Yes

5. Review Comments to the Author

Reviewer #1: This manuscript tested sixteen sarcosine substitution variants for solubility in water. Aqueous soluble variants were then tested in standard complement, myeloperoxidase, NET formation and antioxidant capacity assays. However, we can't see the significance of doing this study, either clinical significance or the structural biological significance.

Reviewer #2: revise the manuscript as their are some missed letters in some words.

Pooled Normal Human Serum (NHS) (add their preparation)

authors should give a list for abbreviations

Figure ligands were written in the result section: they should be separated at the end of the manuscript.

Reviewer #3: The studies in the paper are repeats with altered peptides derived by various sarcosine substitutions of studies of the previously described designed moleceule PA-dPEG24 that the authors have demonstrated have inhibitory effects on a number of mechanisms and mediators of inflammation. In addtion to dermining that 6 of the sarcosine-substituted peptides are soluble, the authors determined their funcational ability to inhibit comlement activation, myeloperoxidase inhibition, NET formation and antioxidant capacity assays in appropriately designed and suitably controlled experiments that, surprisingly, demonstrate that some of the sacosine-substitution peptides perform better than the parent molecule.

The paper is clearly written and easy to follow. The figures are clear.

The potential theraputic uses of the peptides based on their described in vitro activity is not discussed by the authors and this may be appropriate for this paper, but are obvious.

6. PLOS authors have the option to publish the peer review history of their article (what does this mean?). If published, this will include your full peer review and any attached files.

Reviewer #1: No

Reviewer #2: Yes: Rehab Mahmoud Abd El-Baky

Reviewer #3: No

---

## [Author Response · Author response to Decision Letter 0]

1 Dec 2019

Editor: Please confirm that this does not alter your adherence to all PLOS ONE policies on sharing data and materials, by including the following statement: "This does not alter our adherence to PLOS ONE policies on sharing data and materials.” Please include your updated Competing Interests statement in your cover letter; we will change the online submission form on your behalf.

RESPONSE: We have added the requested statement to the cover letter.

Comments to the Author

5. Review Comments to the Author

Reviewer #1: This manuscript tested sixteen sarcosine substitution variants for solubility in water. Aqueous soluble variants were then tested in standard complement, myeloperoxidase, NET formation and antioxidant capacity assays. However, we can't see the significance of doing this study, either clinical significance or the structural biological significance.

RESPONSE: We understand the Reviewer’s concern and have addressed this by expanding the Discussion section with additional details about the clinical significance. (See response to reviewer #3).

Reviewer #2: revise the manuscript as their are some missed letters in some words.

RESPONSE: We performed a spell check and corrected identified deficiencies.

Pooled Normal Human Serum (NHS) (add their preparation)

RESPONSE: We have added the requested description of NHS preparation to the Methods section.

authors should give a list for abbreviations

RESPONSE: We have added a list of abbreviations, although this is not requested in the PLoS One submission guidelines.

Figure ligands were written in the result section: they should be separated at the end of the manuscript.

RESPONSE: We have followed PLoS One submission guidelines which state, “Figure captions must be inserted in the text of the manuscript, immediately following the paragraph in which the figure is first cited.”

Reviewer #3: 

The potential theraputic uses of the peptides based on their described in vitro activity is not discussed by the authors and this may be appropriate for this paper, but are obvious.

RESPONSE: We agree with the Reviewer’s note and have added additional detail to the Discussion about potential therapeutic uses.

---

## [Editor Report · Decision Letter 1]

10 Dec 2019

Inhibition of complement activation, myeloperoxidase, NET formation and oxidant activity by PIC1 peptide variants.

PONE-D-19-26436R1

Dear Dr. Cunnion,

We are pleased to inform you that your manuscript has been judged scientifically suitable for publication and will be formally accepted for publication once it complies with all outstanding technical requirements.

With kind regards,

Yan Li

Academic Editor

PLOS ONE
---

## [Editor Report · Acceptance letter]

18 Dec 2019

PONE-D-19-26436R1 

Inhibition of complement activation, myeloperoxidase, NET formation and oxidant activity by PIC1 peptide variants. 

Dear Dr. Cunnion:

I am pleased to inform you that your manuscript has been deemed suitable for publication in PLOS ONE. Congratulations! Your manuscript is now with our production department. 

With kind regards,

on behalf of

Dr. Yan Li 

Academic Editor

PLOS ONE